# Make Small Data Great Again: Learning from Partially Annotated Data via Policy Gradient for Multi-Label Classification Tasks

## Abstract

Traditional supervised learning methods are heavily reliant on human-annotated datasets. However, obtaining comprehensive human annotations proves challenging in numerous tasks, especially multi-label tasks. Therefore, we investigate the understudied problem of partially annotated multi-label classification. This scenario involves learning from a multi-label dataset where only a subset of positive classes is annotated. Besides the negative classes unknown problem, this task encounters challenges associated with a scarcity of positive class annotations and severe label imbalance. To overcome these challenges, we propose Partially Annotated reinforcement learning with a Policy Gradient algorithm (PAPG), a framework combining the exploration capabilities of reinforcement learning with the exploitation strengths of supervised learning. By introducing local and global rewards to address class imbalance issues and employing an iterative training strategy equipped with data enhancement, our framework showcases its effectiveness and superiority across diverse classification tasks.

## 1 Introduction

Traditional supervised learning methods heavily rely on human-annotated data sets, especially neural approaches that are data-hungry and susceptible to over-fitting when lacking training data. Nevertheless, in some tasks, human annotations are difficult for the needs of specific domain knowledge provided by professional experts. For instance, the task of document-level relation extraction (Yao et al., 2019) seeks to identify meaningful relationships between entity pairs within a document. However, human annotators find it hard to completely annotate all the relations due to the confusion of understanding relation definitions and long-context semantics. This phenomenon of incomplete annotation is likely to exist in many multi-label tasks that generally have dozens, hundreds, or even thousands sizes of class sets (Cole et al., 2021; Tan et al., 2022; Ben-Baruch et al., 2022).

Based on the above observation, we focus on addressing a noteworthy yet under-studied problem, **partially annotated multi-label classification** (PAMLC), which involves learning from a multi-label dataset in which only *a subset of positive classes* is annotated, while all the remaining classes are unknown. Dealing with such a problem faces many challenges: 1) completely unknown negative classes; 2) severe positive-negative imbalance that multi-label classification task faced and missing of positive class annotations exacerbating the imbalance and plaguing recognizing positive classes; 3) both training set and validation set being partially annotated leading to inestimable distribution prior. Figure 1 A exhibits an example from partially annotated relation data, and Figure 1 B illustrates that the entire dataset contains a limited number of positive annotations, with the majority of labels remaining unannotated.

Previous works on multi-label classification with missing labels mainly focus on image classification tasks. Specifically, the settings of multi-label classification with missing labels are roughly divided into three categories: partially observed labeling (POL) which supposes partial positive *and* negative classes are labeled; partially positive observed labeling (PPL) which supposes only part of positive classes are observed but requires at least one positive label per instance; single positive labeling (SPL) which supposes *only* one positive class per instance is observed. The most related setting to us is PPL. However, it assumption of at least one positive label is not suitable for our problem. We consider a more general setting that is not limited to image classification, thus facing tasks

Figure 1: Illustration of PAMLC task. **A.** Partially annotated data samples in document relation extraction. **B.** Severe imbalanced distribution of positive (red scatters) and unannotated classes (blue scatters). **C.** Performance comparison of different approaches for document-level relation extraction.

where an instance may not have any positive classes. For example, it is possible that there is no relation between an entity pair in relation extraction task. For the image classification task, methods designed for these settings could be directly applied to our setting but suffer performance drops due to insufficient necessary information (*i.e.*, partial negative classes or at least one positive class). Another worth mentioning task is Positive-Unlabeled (PU) learning (Jaskie & Spanias, 2019). Different from our setting, traditional PU learning only focuses on learning a *binary* classifier with an incomplete set of positives and a set of unlabeled samples. Besides, many advanced methods of PU learning (Su et al., 2021; Su et al., 2022; Su et al., 2021) generally assume a given positive-negative distribution prior or a balanced positive-negative ratio, which cannot readily adapt to our multi-label settings. We discuss the details of all the settings in the related work section.

There are some simple approaches, *i.e.*, negative mode or re-weight strategies, that could be applied to our problem settings. Negative mode treats all unknown classes as negatives, which learns a biased distribution and causes high precision and low recall evaluation scores on the complete test set. Re-weight strategies based on negative mode generally contain positive up-weight and negative under-weight or sampling (Li et al., 2020). We attempt these approaches and observe that the performance of the negative mode drops significantly when the ratio of annotated positive classes decreases. Re-weight strategies partly improve the model performance but still perform unsatisfactorily when only a very small set (10%) of positive class annotations is available (shown in Figure 1 C).

Previous works (Silver et al., 2016; Feng et al., 2018; Nooralahzadeh et al., 2019) have demonstrated the powerful exploration ability of Reinforcement Learning (RL). Furthermore, RL has been shown great success on distant or even zero annotations. To deal with the partially annotated multi-label classification problem with imbalanced issues, we propose an RL-based framework, namely Partially Annotated Policy Gradient (PAPG), that is devoted to estimating an unbiased distribution only with the observation of annotated labels. Specifically, we combine the exploration ability of reinforcement learning and the exploitation ability of supervised learning by designing a policy network (as a multi-label classifier) learning with a policy-based RL algorithm and a value network (as a critic) trained with a supervised loss to provide local reward. Besides, beyond the local reward, we design a global reward assessing predictions of each instance, which contributes most to alleviating the imbalanced problem. In addition, inspired by the actor-critic RL algorithm (Bahdanau et al., 2016), we iteratively train the value network and the policy network, which achieves dynamic reward estimation and a one-stage training procedure. To gradually enhance the assessment accuracy of our value network, we carefully select predicted positive classes with high confidence to enhance the training set of the value network.

Moreover, our RL framework is concise and flexible to guarantee its generalization and adaptation to many tasks. We conduct sufficient experiments across various complicated domains: synthetic toy setting (§4.1), multi-label image classification task (§4.2), and multi-relation document extraction task (§4.3). All experimental results show the effectiveness of our framework and demonstrate a significant improvement over previous work.

## 2 RELATED WORK

**Problem Settings on Weak Annotation Learning**    Weakly supervised learning has long attracted researchers' interest because large-scale datasets with high-quality human annotations are time-consuming and difficult. Weakly supervised learning includes different settings according to different assumptions Among them, noisy-label learning (Coscia & Neffke, 2017; Yang et al., 2018;

Feng et al., 2018; Ren et al., 2020; Cai et al., 2022) typically utilize datasets containing label noise, encompassing instances of both false positives and false negatives; partial-label learning (Zhang & Yu, 2015; Xu et al., 2019; Wang et al., 2021; Wu et al., 2022; Tian et al., 2023) refers to the task where each training instance is annotated with a set of candidate labels in which only one is the target; semi-supervised learning (Berthelot et al., 2019; Zheng et al., 2022; Yang et al., 2022) typically leverages a small set of full-labeled data and an amount of unlabeled data; PU learning (Kiryo et al., 2017; Su et al., 2021; Acharya et al., 2022) typically refers to learning a binary classifier (positive vs. negative) using an incomplete set of positives and a set of unlabeled samples without any explicitly labeled negative examples. Our problem setting belongs to the domain of weak-supervised learning but distinguishes itself from all preceding problem settings.

**Learning Methods for Partial Supervision** Similar to our setting, some previous work addresses partial supervision problems. There are some methods, such as partial conditional random field, that deal with single-label multi-class tasks with partial supervision (Mayhew et al., 2019; Effland & Collins, 2021; Li et al., 2021; Zhou et al., 2022). In the multi-label classification area, Cole et al. (2021); Kim et al. (2022) assumes a specific setting where only one label for each image is available. Ben-Baruch et al. (2022) designs a class-selective loss to tackle the partial observation of a small set of positives *and* negatives. Similarly, Durand et al. (2019); Durand et al. (2022) also suppose partial observation of positive and negative classes. The methods of PU learning contain cost-sensitive approaches (Christoffel et al., 2016; Su et al., 2021) which assume the data distribution prior to achieving unbiased risk estimation; representation clustering approaches Acharya et al. (2022) leveraging contrastive learning to generate pseudo-positive/negative labels; and sample-selection approaches (Zhang & Zuo, 2009; Luo et al., 2021) which are devoted to finding likely negatives from the unlabeled data according to the heuristic methods or sample confidence. These methods cannot be effectively adapted to our task, because it is impractical to know or estimate the label distribution prior to the multi-label dataset in our setting where all the negative labels are missing. We note a concurrent work (Yuan et al., 2023) extending positive-unlabeled learning into a multi-label classification task. Nevertheless, their research has primarily concentrated on method development within the domain of image classification. In contrast, our framework exhibits broader applicability, extending its utility to various tasks.

**Reinforcement Learning under Weak Supervision** There are many previous works leveraging Reinforcement Learning (RL) to solve tasks only with weak supervision (Feng et al., 2018; Zeng et al., 2018; Luo et al., 2021; Chen et al., 2023). In the NLP field, to precisely leverage distant data, Qin et al. (2018); Feng et al. (2018) train an agent as a noisy-sentence filter, taking performance variation on development or probabilities of selected samples as a reward and adopting policy gradient to update. Nooralahzadeh et al. (2019) expand their methods to NER task. Recent work of Chen et al. (2023) also conducts RL to remove the noisy sentence so as to improve the fault diagnosis system. Notably, RL learning on distantly supervised learning aims to filter false positives, whereas our goal is to identify false negatives. A closer related work to us is Luo et al. (2021), in which an RL method is designed to solve the PU learning problem. But unlike us, their agent is a *negative sample selector*, aiming to find negatives with high confidence and then couple them with partial positives to train a classifier. Besides, they suppose a fully annotated validation dataset and a balanced positive-negative distribution prior.

## 3 REINFORCEMENT LEARNING WITH PARTIAL SUPERVISION

We propose a new RL framework to solve the partially annotated multi-label classification (PAMLC) task. We formulate the multi-label prediction as the action execution in the Markov Decision Process (MDP) (Puterman, 1990). We design both *local* and *global* rewards for actions to guide the action decision process. The policy-based RL method is adopted to train our policy network. We introduce the details of our RL framework in the following subsections.

### 3.1 PROBLEM SETTING

We first mathematically formulate the PAMLC task. Given a multi-label dataset $\mathcal{X} = \{\mathbf{x}_i\}$, each $\mathbf{x}_i \in \mathcal{X}$ is labeled with a partially annotated multi-hot vector $\mathbf{y}_i = [y_i^1, \cdots, y_i^c, \cdots, y_i^{|\mathcal{C}|}]$, where $y_i^c \in \{0, 1\}$ denotes whether class $c \in \mathcal{C}$ is TRUE for instance $\mathbf{x}_i$ and $|\mathcal{C}|$ is the cardinality of set $\mathcal{C}$. In terms of partial annotation, we assume that $\{c; y_i^c = 1\}$ is a subset of gold positive classes regarding to $\mathbf{x}_i$, *i.e.*, $\{c; y_i^c = 0\}$ is the set of UNKNOWN classes. Typically, in the multi-label setting,

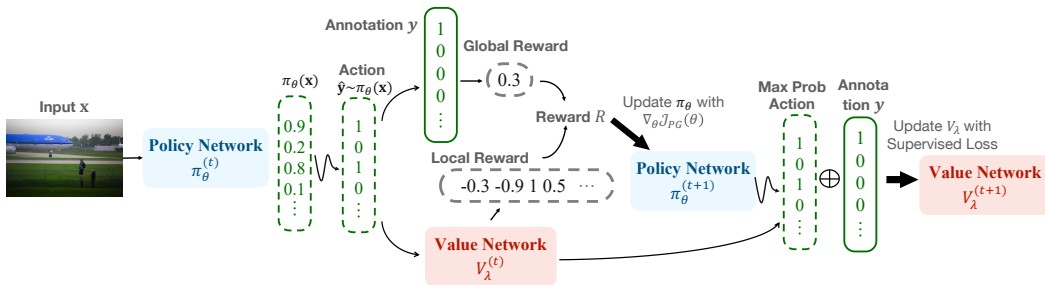

Figure 2: Illustration of our RL framework. $\oplus$ represents union operation. We iteratively update the policy network and value network. The augmented training data are curated for the value network.

the size of the label set is dozens or hundreds; thus $|\{c; y_i^c = 1\}| << |\{c; y_i^c = 0\}| \leq |\mathcal{C}|$. In some complicated tasks, such as Relation Extraction, we further define a special label `<None>` to $\mathbf{x}_i$ if $\mathbf{y}_i = [y_i^c = 0]_{c=1}^{|\mathcal{C}|}$. It should be mentioned that we do not have any fully annotated data, both the training set and validation set being partially annotated.

As aforementioned, negative mode approach is susceptible to training models that conform to the imbalanced data distribution, resulting in high precision but low recall on the test set. Based on the negative mode, we introduce a reinforcement learning (RL) framework designed to mitigate distribution bias and encourage the multi-label classifier to predict more potential positive classes.

### 3.2 MODELING

Typically, basic RL is modeled as an MDP $(S, A, \pi, \mathcal{T}, R)$ which contains a set of environment and agent states $S$, a set of actions $A$ of the agent, the transition probabilities $\mathcal{T}$ from a state to another state under action $a$, and the reward $R$. The goal of an RL agent is to learn a policy $\pi$ that maximizes the expected cumulative reward. In our problem setting, we do not take state transitions into account because our action execution does not change the agent and environment. The policy $\pi_\theta$ of our RL framework is a multi-label classifier constructed by a neural network with parameter $\theta$. We following define the constituents in our RL framework in detail.

**States**  A state $s$ includes the potential information of an instance to be labeled. In our setting, this information consists of instance features, which are essentially continuous real-valued vectors derived from a neural network.

**Actions**  Due to the multi-label setting, our agent is required to determine the label of each class $c$ for one instance. There are two actions for our agent: setting the current class as TRUE ($y_i^c = 1$) or UNKNOWN ($y_i^c = 0$). Consequently, it is necessary to execute a sequence of (size of class set) actions to completely label an instance.

**Policy**  Our policy network outputs the probability $\pi_\theta(y_i^c | \mathbf{x}_i) = P(a = y_i^c | s = \mathbf{x}_i)$ for each action condition on the current state. we adopt the model structure commonly utilized in previous supervised studies as the architecture for our policy network.

**Rewards**  Recall that our primary objective is to acquire a less biased label distribution compared to the supervised negative mode training approach using the partially annotated training dataset. We anticipate that our PAPG possesses the capacity for balanced consideration of both *exploitation* and *exploration*. *exploitation* ensures that our agent avoids straying from local optima direction, while *exploration* motivates our agent to adapt its policy, preventing overfitting to partial supervision within the broader global context. Inspired by the actor-critic RL algorithm (Bahdanau et al., 2016), we design our rewards function containing two parts: a **local reward** from a trainable value network, which provides immediate value estimation of each action and a **global reward** regarding the overall performance of the whole actions sequence for each instance.

Specifically, the local reward calculates the reward of each action (for each class) of each instance according to the value network confidence to an action:

$$r_i^c(V_\lambda, \mathbf{x}_i, c) = \begin{cases} \mathbb{C}(-1, \log \frac{p_{V_\lambda}^c(\mathbf{x}_i)}{1 - p_{V_\lambda}^c(\mathbf{x}_i)}, 1) & \text{if } \hat{y}_i^c = 1, \\ \mathbb{C}(-1, \log \frac{1 - p_{V_\lambda}^c(\mathbf{x}_i)}{p_{V_\lambda}^c(\mathbf{x}_i)}, 1) & \text{if } \hat{y}_i^c = 0. \end{cases} \tag{1}$$

where $p_{V_\lambda}^c(\mathbf{x}_i)$ is the probability of class $c$ being TRUE for instance $\mathbf{x}_i$, calculated by a value network $V_\lambda$, and $\mathbb{C}(-1, \cdot, 1)$ is a clamping function: a) $\mathbb{C}(-1, x, 1) = -1$ if $x < -1$; b) $\mathbb{C}(-1, x, 1) = 1$ if $x > 1$; c) otherwise, $\mathbb{C}(-1, x, 1) = x$. We sum out the local rewards of all classes as part of the final rewards of an instance.

Intuitively, the local rewards offer a preliminary yet instructive signal to guide the learning process in our PAPG framework. This signal inherits the *exploitation* aspect from the supervised loss training (as the value network is trained through supervised learning). Its purpose is to prevent the PAPG from engaging in excessively invalid exploratory behavior within the large action space, thereby enhancing the overall learning efficiency. Nevertheless, relying solely on these local rewards may potentially lead the PAPG system to converge to a biased "negative mode" solution. To mitigate this risk, we introduce global rewards to stimulate more comprehensive exploration during the learning process.

As for the *global* reward, we employ a straightforward yet highly effective scoring function, which is computed based on the recall metric. In detail, for the whole classes prediction $\hat{\mathbf{y}}_i$ of $\mathbf{x}_i$ with the observed ground truth $\mathbf{y}_i$, the recall score is:

$$recall(\mathbf{y}_i, \hat{\mathbf{y}}_i) = \frac{|\{y_i^c = 1 \wedge \hat{y}_i^c = 1, \ y_i^c \in \mathbf{y}_i, \hat{y}_i^c \in \hat{\mathbf{y}}_i\}|}{|\{y_i^c = 1, \ y_i^c \in \mathbf{y}_i\}|} \tag{2}$$

The key insight here revolves around the "recall" metric, which serves as an exactly accurate evaluation measure (compared to "precision" and "F1") because a substantial portion of positive instances remains unlabeled within the context of partially annotated datasets. Furthermore, to enhance recall scores, the policy network is encouraged to predict a greater number of classes as TRUE.

The complete reward function for a sampled action sequence of an instance is:

$$R(\mathbf{x}_i, \hat{\mathbf{y}}_i, V_\lambda, \mathbf{y}_i) = \frac{1}{|\mathcal{C}|} \sum_{c \in \mathcal{C}} r_i^c(V_\lambda, \mathbf{x}_i, c) + w * recall(\mathbf{y}_i, \hat{\mathbf{y}}_i), \tag{3}$$

where $w$ is a weight controlling the scale balance between local reward and global reward.

### 3.3 PREDICTION

The final predictions of each instance are decided according to the policy network. Classes whose probabilities are more than $0.5$ ($\pi_\theta(y_i^c|\mathbf{x}_i) > 0.5$) are set to as TRUE. Note that our value network can directly output the prediction of each class for an instance. However, we only leverage its predictions to label enhancement for the reason of reducing the influence of enhanced noisy labels.

### 3.4 LEARNING

We iteratively train our value network and policy network in an end-to-end fashion. Since the value network plays a critical role in guiding policy network learning, we employ data enhancement techniques during the training of the value network to enhance the precision of value estimations. It is important to emphasize that we intentionally exclude the enhanced data from participation in the calculation of the recall reward. This decision is motivated by the desire to maintain the precision of the global reward and prevent potential noise introduced by the enhanced data. Additionally, it is noteworthy that the enhanced data could contribute to the benefit of the policy network through the local reward signal provided by the value network.

It is widely acknowledged that the training process in RL can experience slow convergence when confronted with a vast exploration space. Inspired by previous RL-related works (Silver et al., 2016; Qin et al., 2018), we initiate our process by conducting pre-training for both our policy and value networks before proceeding with the RL phase. Typically, pre-training is executed through a supervised method. In our settings, a range of trivial solutions for PAMLC can serve as suitable candidates for pre-training. We simply utilize the negative mode for pre-training in most cases. However, as previously mentioned, the negative mode has a tendency to acquire a biased label distribution due to the presence of severe label imbalance. Thus, we implement an early-stopping strategy during the pre-training phase to prevent convergence. We following introduce the detailed learning strategies and objectives.

**Objective for Value Model**    Generally, a well-designed supervised objective urges models to learn expected outputs by learning from annotated data. This process typically refers to the *exploitation*, where the supervised model fits the distribution of label annotations. We denote the supervised

objective by a general formulation:

$$\mathcal{L}_{SUP}(\theta) = \sum_{\mathbf{x}_i \in \mathcal{X}} p(\mathbf{x}_i)\mathcal{D}(\mathbf{y}_i, \hat{\mathbf{y}}_i), \tag{4}$$

where $\mathcal{D}$ is a task-specific distance metric measuring the distance between annotation $\mathbf{y}_i$ and prediction $\hat{\mathbf{y}}_i$. Recall that we treat all the unannotated classes as negatives in the partially annotated setting in order to perform supervised learning.

**Objective for Policy Model**    As stated in previous work (Qin et al., 2018), policy-based RL is more effective than value-based RL in classification tasks because the stochastic policies of the policy network are capable of preventing the agent from getting stuck in an intermediate state. We leverage policy-based optimization for RL training. The objective is to maximize the expected reward:

$$\mathcal{J}_{PG}(\theta) = \mathbb{E}_{\pi_\theta}[R(\theta)] \approx \sum_{\mathbf{x}_i \in batch} p(\mathbf{x}_i) \sum_{\hat{\mathbf{y}}_i \sim \pi_\theta(\hat{\mathbf{y}}_i|\mathbf{x}_i)} \pi_\theta(\hat{\mathbf{y}}_i|\mathbf{x}_i)R(\hat{\mathbf{y}}_i.\mathbf{x}_i), \tag{5}$$

The policy network $\pi_\theta$ can be optimized w.r.t. the policy gradient REINFORCE algorithm (Williams, 1992), where the gradient is computed by

$$\nabla_\theta \mathcal{J}_{PG}(\theta) = \sum_{\mathbf{x}_i \in batch} p(\mathbf{x}_i) \sum_{\hat{\mathbf{y}} \sim \pi_\theta(\hat{\mathbf{y}}|\mathbf{x}_i)} \nabla_\theta \ln(\pi_\theta(\hat{\mathbf{y}}|\mathbf{x}_i))R(\hat{\mathbf{y}}, \mathbf{x}_i), \tag{6}$$

where $p(\mathbf{x}_i)$ is a prior distribution of input data. Specific to uniform distribution, $p(\mathbf{x}) = \frac{1}{|\mathbf{x}_{batch}|}$.

**Overall Training Procedure**    The overall training process is demonstrated in Algorithm 1, where $\hat{y}_i^c = V_\lambda(\mathbf{x}_i)_c = 1$ refers to the prediction of class $c$ that value network outputs for the sample $\mathbf{x}_i$ being TRUE. It needs to be clarified that our computation of local rewards for actions is based on the annotated positive classes and a randomly selected subset of unknown classes rather than considering the entire sequence of actions of an instance, which is intended to emphasize the impact of positive classes within the computation of local rewards. We empirically prove the effectiveness of this strategy. Furthermore, we fix the value network once it converges, a strategy employed to enhance training efficiency.

---

**Algorithm 1:** Partially Annotated Policy Gradient Algorithm

---

**Input:** Observed data $\mathcal{X} = \{\mathbf{x}_i\}$, partial annotations $\mathcal{Y} = \{\mathbf{y}_i\}$, pre-trained policy network $\pi_{\theta^0}$, value network $V_\lambda$, REINFORCE learning rate $\alpha$, confidence threshold $\gamma$

**Output:** Optimal parameters $\theta^*$

1  $e \leftarrow 0, \theta^* \leftarrow \theta^0$, enhanced annotation set $\bar{\mathcal{Y}} \leftarrow \mathcal{Y}$

2  **while** $e <$ *total training epoches* **do**

3      Training set for value network: $(\mathcal{X}, \bar{\mathcal{Y}})$

4      Training set for policy network: $(\mathcal{X}, \mathcal{Y})$

5      **for** $\mathcal{X}_{batch} \in \mathcal{X}$ *in total batches* **do**

6          Update $\lambda$ by minimizing Equation (4) with $\bar{\mathcal{Y}}_{batch}$ for value network

7          **for** *step* $t <$ *sample steps* $T$ **do**

8              For each $\mathbf{x}_i \in \mathcal{X}_{batch}$, sample $\hat{\mathbf{y}}_i$ w.r.t. $\hat{\mathbf{y}}_i \sim \pi_\theta(\hat{\mathbf{y}}_i|\mathbf{x}_i)$

9              Compute $R(\mathbf{x}_i, \hat{\mathbf{y}}_i, V_\lambda, \mathbf{y}_i)$ according to Equation (3)

10         update policy network using $\theta \leftarrow \theta + \alpha\nabla_\theta\mathcal{J}_{PG}(\theta)$ according to Equation (6)

11     $\bar{\mathcal{Y}} \leftarrow \{[\bar{y}_i^c]_{c=1}^{|\mathcal{C}|}\}$, where $\bar{y}_i^c = 1$ if ($y_i^c = 1$ or ($\pi_\theta(\hat{y}_i^c = 1|\mathbf{x}_i) > \gamma$ and $\hat{y}_i^c = V_\lambda(\mathbf{x}_i)_c = 1$))

12     **if** *eval*($\pi_\theta$) $>$ *eval*($\pi_{\theta^*}$) **then**

13         $\theta^* \leftarrow \theta$

14     $e \leftarrow e + 1$

15 **return** $\theta^*$

---

## 4   EXPERIMENTS

To verify the effectiveness of our proposed RL framework, we experiment with three widely-concerned tasks. The first one follows the general setting in PU learning, but we consider an im-

Table 1: F1 scores with varying ratios of positive annotations. We take images of the "Airplane" category as positives in this table.

| Method | 10% | 20% | 30% | 40% | 50% | 60% | 70% | 80% | 90% |
|---|---|---|---|---|---|---|---|---|---|
| nnPU | 44.8 | 47.9 | 49.1 | 50.4 | 52.8 | 54.1 | 56.1 | 57.4 | 56.7 |
| ImbalancednnPU | 48.6 | 53.0 | 59.1 | 62.8 | 65.2 | 64.6 | 64.9 | 67.7 | 69.2 |
| Negative Mode | 7.0 | 17.1 | 28.7 | 37.6 | 56.9 | 55.6 | 64.2 | 71.2 | 75.2 |
| PAPG (Ours) | 50.2 | 64.6 | 66.6 | 67.8 | 69.8 | 70.8 | 75.1 | 75.8 | 76.9 |

balanced situation. This experiment aims to prove the adaptation of our framework to binary classification problems. The second and the third focus on two partially annotated multi-label tasks, document-level relation extraction and multi-label image classification, selected from classical Natural Language Processing (NLP) tasks and Computer Vision (CV) tasks, respectively.

### 4.1 EXP1: SYNTHETIC CLASSIFICATION

We conduct binary image classification experiments with the same setting following Su et al. (2021) that concentrates on positive/negative imbalanced problems in PU learning. With our formulation, an instance $\mathbf{x}_i$ is an image, and the label of an instance in binary classification settings can be denoted as $\mathbf{y}_i = [y_i^1, y_i^2]$ where $y_i^1$ is the label of positive and $y_i^2$ is the label of negative. The prediction for each image is conducted by setting $y$ corresponding to the higher score as 1 and the other as 0.

**Dataset** The imbalanced datasets are constructed from CIFAR10[1] by picking only one category as positives and treating all other data as negatives. Hence, there are 50,000 training data and 10,000 test data as provided by the original CIFAR10. To make the training data into a partially annotated learning problem, we randomly sample a ratio of positives as annotated data and all the leaving training data as an unknown set.

**Configuration and Baselines** Of note, our framework can integrate any supervised model architecture. For a fair comparison, we take the same architecture of Kiryo et al. (2017); Su et al. (2021) as our value and policy networks, *i.e.*, a 13-layer CNN with ReLU and Adam as the optimizer. Kiryo et al. (2017)

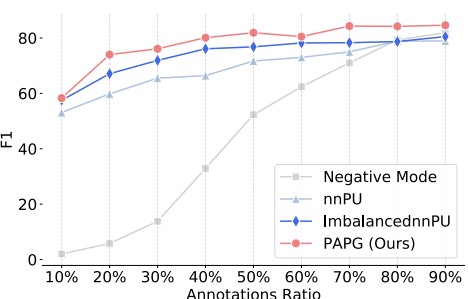

Figure 3: Experimental results of the setting with "truck" category as positives.

designed an algorithm **nnPU** for PU learning with balanced binary classification data, while Su et al. (2021) proposed **ImbalancednnPU** considering imbalanced setting. We take these two previous state-of-the-art models as our compared baselines. We re-run **nnPU** and **ImbalancennPU** with their provided codes and configurations and report the results. We tune the hyper-parameter $w$ in Eq.3 between $\{10, 20, 50\}$ with different experiment settings, and $w$ is dynamically adjusted during training[2]. The action sampling number $T$ is 100, and the negative reward sampling is 20% for all settings. The threshold $\gamma$ to choose enhancement labels is 0.8. We keep the values of other hyper-parameters the same as Su et al. (2021). Following previous work, we evaluate all the methods with F1 scores. Unless stated otherwise, the hyper-parameters specified in our framework remain the same in the following experiments.

**Results** We show F1 scores with varying ratios of annotated positives in Table 1 (Precision and Recall metrics can be found in Appendix B.2). We perform supervised training with Binary Cross-Entropy (BCE) loss in **Negative Mode**. Naturally, the supervised training performs worse by worse as the ratios of annotated positives decrease. It is obvious that **Our PAPG** achieves significant improvements over **Negative Mode**, especially when the ratio of annotated data becomes lower. Compared to previous work, our framework still demonstrates its superiority even though **ImbalancednnPU** is specifically designed for PU learning with imbalanced settings. Note that according

---

[1]A multi-class dataset containing ten categories. https://web.cs.toronto.edu/

[2]Intuitively, the $w$ of recall reward should be dropped along with the training epochs because our value network provides more and more accurate local rewards beneficial by data enhancement before convergence.

Table 2: Experimental Results on COCO datasets with varying ratios of positive classes annotations.

| Method | 10% | | 30% | | 50% | | 70% | | 90% | |
|---|---|---|---|---|---|---|---|---|---|---|
| | F1 | mAP | F1 | mAP | F1 | mAP | F1 | mAP | F1 | mAP |
| ERP | - | 63.8 | - | 71.0 | - | 73.5 | - | 73.8 | - | 74.4 |
| ROLE | - | 58.2 | - | 72.4 | - | 76.6 | - | 79.5 | - | 81.1 |
| P-ASL+Negative | 45.2 | 66.9 | 52.1 | 74.6 | 54.0 | 76.9 | 71.9 | 81.0 | 80.3 | 83.3 |
| P-ASL+Counting | 5.1 | 46.4 | 26.4 | 63.4 | 53.7 | 76.1 | 71.6 | 80.1 | 60.4 | 80.4 |
| Negative Mode | 6.4 | 50.6 | 33.7 | 64.3 | 52.9 | 73.8 | 72.3 | 81.2 | 80.1 | 83.5 |
| PAPG (Ours) | 68.3 | 66.6 | 77.0 | 77.5 | 79.1 | 80.4 | 79.0 | 81.4 | 80.5 | 83.4 |

to the data construction, any category of data can be chosen as the positive set. To further make our experiments convincing, we show the results of different data construction in Figure 3.

## 4.2 EXP2: MULTI-LABEL IMAGE CLASSIFICATION

We demonstrate the effectiveness of our method on the multi-label image classification task.

**Dataset** Following Ben-Baruch et al. (2022) (**P-ASL**), which deals with partial annotations containing both positive and negative classes, we utilize MS-COCO dataset (Lin et al., 2014) containing 80 classes. We keep the original split with 82081 training samples, and 40137 test samples. We simulate the partial annotations following the operations in **P-ASL**. But different from them, we only retain the positive classes in their partial annotations and take all the rest of the classes as UNKNOWN.

**Configuration and Baselines** For a fair comparison, our value and policy networks have the same architecture as **P-ASL**. Due to the different partially annotated settings, we re-run **P-ASL** utilizing their codebase but with our datasets. **P-ASL+Negative** means training a model taking all UNKNOWN as negative classes to predict label distribution as prior. **P-ASL+Counting** means counting partially labeled positive classes as distribution prior. We also re-run **EPR** and **ROLE** methods from Cole et al. (2021) with our datasets, utilizing their official code. We tune the hyper-parameter $w$ between $\{5, 7, 12\}$ in this task. Following previous work (Ridnik et al., 2021; Huynh & Elhamifar, 2020), we use both F1 scores and mAP as evaluation metrics in this task. Detailed methodology of the re-weight approach and the detailed formula of metric calculations can be found in Appendix A.

**Results** Experimental results with different annotation ratios are shown in Table 12. We can find that without annotated negative classes to estimate a relatively accurate distribution prior, **P-ASL** performs unsatisfactorily, especially in F1 scores when the annotation ratios decrease. **Our PAPG** model outperforms all baselines in all settings. Furthermore, we ran our model three times and found very small standard deviations of F1 scores and mAP, which demonstrates the high robustness and stability of our framework. Standard deviations of three runs and more experimental results can be found in Appendix B.2.

## 4.3 EXP3: DOCUMENT-LEVEL RELATION EXTRACTION

Document-level Relation Extraction (DocRE) is a task that focuses on extracting fine-grained relations between entity pairs within a lengthy context. Align to our formulation, an input $\mathbf{x}_i$ is an entity pair, and $\mathbf{y}_i$ represents relation types between the entity pair. An entity pair may have multiple relations or have no relation in DocRE. Thus, we conduct this experiment to study how our framework performs when <None> label exists in a multi-label classification task.

**Dataset** We chose the Re-DocRED which is the most complete annotated dataset in DocRE. The size of label set $\mathcal{C}$ is 97 (contains <None>) in Re-DocRED. To simulate partial annotation, we randomly kept a ratio of annotated relations as the above experiments do.

**Configuration and Baselines** In this experiment, we adopt the state-of-the-art model of DREEAM (Ma et al., 2023) as the architectures of our value and policy networks. Meanwhile, we keep the same training method with an Adaptive Thresholding Loss (ATL) of DREEAM for our value network. To the best of our knowledge, there are currently no existing methods specifically designed for addressing partially annotated DocRE, so we only take **Pos Weight** and **Neg Weight**, adopting the same methodologies as the image classification experiments, as our compared baselines. The threshold $\gamma$ of choosing enhancement labels is 0.95, and we find our framework performs robust to $\gamma$ varying from 0.5 to 0.95.

Table 3: Results on Re-DocRED datasets with varying ratios of positive classes annotations

| Method | **10%** | | | **30%** | | | **50%** | | | **70%** | | | **100%** | | |
|---|---|---|---|---|---|---|---|---|---|---|---|---|---|---|---|
| | P | R | F1 | P | R | F1 | P | R | F1 | P | R | F1 | P | R | F1 |
| **DREEAM** | 89.8 | 3.8 | 7.3 | 92.0 | 20.8 | 33.8 | 91.8 | 42.1 | 57.7 | 89.6 | 58.8 | 70.6 | 86.0 | 72.4 | 78.6 |
| **Pos Weight** | 84.9 | 39.8 | 54.1 | 85.5 | 59.3 | 70.0 | 85.0 | 66.8 | 74.8 | 83.4 | 72.5 | 77.6 | 82.9 | 77.3 | 80.0 |
| **Neg Weight** | 86.7 | 30.7 | 45.4 | 85.0 | 59.3 | 69.8 | 84.1 | 68.2 | 75.3 | 82.8 | 72.8 | 77.5 | 79.8 | 78.7 | 79.3 |
| **PAPG (Ours)** | 58.5 | 77.0 | 66.0 | 83.5 | 67.71 | 74.7 | 81.4 | 73.6 | 77.3 | 83.3 | 73.9 | 78.3 | 80.9 | 80.8 | 80.9 |

Table 4: Ablation study on our rewards.

| Method | **Re-DocRED** | | | **COCO** | | |
|---|---|---|---|---|---|---|
| | P | R | F1 | P | R | F1 |
| **PAPG (Ours)** | 64.5 | 72.8 | 68.4 | 80.9 | 59.2 | 68.3 |
| w/o. Local reward | 12.2 | 93.3 | 21.6 | 61.5 | 65.5 | 63.4 |
| w/o. Global reward | 84.5 | 45.9 | 59.5 | 89.7 | 6.9 | 12.8 |
| w. Prec | 85.9 | 44.0 | 58.1 | 96.2 | 29.8 | 45.5 |
| w. F1 | 86.0 | 43.3 | 57.6 | 89.2 | 46.8 | 61.4 |

Table 5: Ablation study on our training strategy.

| Method | **Re-DocRED** | | | **COCO** | | |
|---|---|---|---|---|---|---|
| | P | R | F1 | P | R | F1 |
| **PAPG (Ours)** | 64.5 | 72.8 | 68.4 | 80.9 | 59.2 | 68.3 |
| w/o. Iterative training | 89.9 | 34.6 | 49.9 | 76.4 | 33.8 | 46.9 |
| w/o. Label enhancement | 83.6 | 47.2 | 60.3 | 51.7 | 54.8 | 53.2 |
| w/o. Action sampling | 88.2 | 36.5 | 51.7 | 96.4 | 20.4 | 33.7 |
| Supervised self-training | 68.0 | 29.0 | 40.7 | 89.7 | 6.6 | 12.3 |

**Results**   Experimental results in Re-DocRED are shown in Table 3. We randomly sample three different versions of datasets and report the average results over them. Similar to **Negative Mode**, **DREEAM** performs supervised training with all unknown classes as negatives in the partially annotated settings, which is our base model in this task. Similar to the above experiments, **Our PAPG** demonstrates its advantage in all annotation ratios. It is worth noting that our framework also achieves improved performance with the full annotated dataset because the full annotations of Re-DocRED still miss some actual relations as aforementioned in the introduction. We also show precision and recall evaluation metrics in this task. It can be seen that our framework achieves consistent improvement in recall scores, suggesting its ability to deal with imbalanced problems and predict more positive labels. We provide more detailed experimental results in Appendix B.1.

## 4.4    ANALYSIS

We conduct ablation studies to analyze our PAPG framework both on modeling and training strategy.

**Rewards Design:**   To show the effectiveness of combining *exploitation* and *exploration* and the benefit of *local* and *global* rewards, we train our framework in the $10\%$ annotations setting without local rewards and global rewards, respectively. Additionally, we replace the recall scores with precision **w. Prec** or F1 scores **w. F1** as our global rewards to show the effects of different global reward designs. Experimental results are shown in Tabel 4. It can be observed that it is hard for an RL framework to achieve comparable performance without local rewards to guide exploitation. The reason is that the action space of multi-label classification is too large to find the global optimal directions. Without our global reward, the recall evaluation score drops a lot (72.78 vs. 45.94), which demonstrates the big advantage of the global reward in alleviating imbalance distribution. Moreover, both the two variants of global reward damage the performance, revealing the advance of taking the exactly accurate evaluation as rewards in the partially annotated setting.

**Training Strategy:**   To verify the effectiveness of our training procedure, we attempt different training strategies shown in Tabel 5. **w/o. Iterative training** means that we fix the value network after pretraining and only train the policy network in the RL training procedure. **w/o. Data enhancement** means that we still iteratively train our value and policy network but do not enhance pseudo labels for the value network. **w/o. Action sampling** means that we leverage the whole action sequence to calculate local rewards without sampling operation illustrated in Section 3.4. **Supervised self-training** means that we conduct self-training of the value network. It is obvious that our training method makes remarkable achievements. More analysis experiments are in Appendix B.1.

## 5    CONCLUSION

In this work, we propose an RL framework to deal with partially annotated multi-label classification tasks. We design local rewards assessed by a value network and global rewards assessed by recall functions to guide the learning of our policy network, achieving both exploitation and exploration. With an iterative training procedure and a cautious data enhancement, our PAPG has demonstrated its effectiveness and superiority on different tasks in different domains.

## REPRODUCIBILITY STATEMENT

The detailed descriptions of the three experiments covered in this article are described in the configuration of the corresponding subsections in Section 4, and the implementation details of the **Pos Weight** and **Neg Weight** methods are introduced in Appendix A.1. The Dataset module of each subsection clearly describes the construction of the data set used in the experiment. For experiments on hyperparameter selection, see Appendix B.1. All dataset we used are publicly accessible. Codes that can reproduce our results are available at https://anonymous.4open.science/r/iclr2024-64EB/.

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
