# Appendices

## A   MORE TECHNOLOGY DETAILS

### A.1   POS WEIGHT AND NEG WEIGHT

In the "Pos Weight" method, we impose a large weight $w_p$ to the positives. We set $w_p$ as the times of positive targets to unlabeled targets in each training batch. Previous study (Li et al., 2020) had stated that negative sampling can be considered as a type of negative weighting method. And this work experimentally find that negative sampling even work better. In our experiments, we under-sampling the unlabeled targets as the "Neg Weight" method. Unlabeled targets 10 times the number of positive targets are retained in each training batch.

### A.2   EVALUATION METRICS

We compute the F1 scores based on TP (True Positive), FP (False Positive), and FN (False Negative).

$$
\begin{aligned}
Recall &= TP/(TP + FN), \\
Precision &= TP/(TP + FP), \\
F1 &= 2 * Recall * Precision/(Recall + Precision).
\end{aligned}
\tag{7}
$$

We choose the widely used evaluation metric mAP on multi-label image classification. $N_c$ is the number of images containing class $c$, Precision$(k, c)$ is the precision for class $c$ when retrieving $k$ best predictions and rel$(k, c)$ is the relevance indicator function that is 1 if the class $c$ is in the ground-truth of the image at rank $k$. We also compute the performance across all classes using mean average precision (mAP), where $C$ is the number of classes.

$$
\begin{aligned}
AP_c &= \frac{1}{N_c} \sum_{k=1}^{N} Precision(k, c) * rel(k, c), \\
mAP &= \frac{1}{C} \sum_{c} AP_c
\end{aligned}
\tag{8}
$$

## B   MORE EXPERIMENTS

### B.1   DOCUMENT-LEVEL RELATION EXTRACTION

**Training curve**. In Figure 4, we display the reward and loss curves of our model in three annotation rations, 10%, 50%, and 100%. Our experimental settings were conducted under partially annotated multi-label tasks, but we also compute metrics on ground Truth during the experiment. As shown in Figure 5, we take annotations ratio=50% as an example, although the F1 scores were low on the partially annotated dataset, the F1 scores are about 20 percentage points higher on the ground truth.

**All experiments on different ratios of annotated labels** To fully verify the effectiveness and robustness of our model, we randomly constructed three versions of data sets and tested the DREEAM model, Pos Weight, Neg Weight, and our PAPG model on all data sets respectively. The results are shown in Table 6.

**Experiments on selecting action sampling ratios** (Take annotations ratio=50% as an example) In order to select the action sampling ratio hyperparameter, we conducted comparative experiments from 0.1 to 0.9, and finally found that the model performed best when the hyperparameter was 0.4. The results are shown in Table 7.

**Value Network Performance of Our PAPG** We iteratively train our value network and policy network. After multiple rounds of iterations, the performance of value network has been greatly improved. The performance of value network of our PAPG are shown in Table 8.

**Case study**. In Table 13, we show an example on the prediction of each method. Our PAPG predicts more true positives.

## B.2 Multi-label Image Classification

**The experimental results on extra evaluation metrics and annotation ratios**. In Table 9 and Table 10, we show Precision and Recall of CIFAR10 and the results of other annotation ratios on Ms-COCO. Table 11 shows the stability of our method PAPG. The standard deviation was computed from three different runs on the MS-COCO dataset.

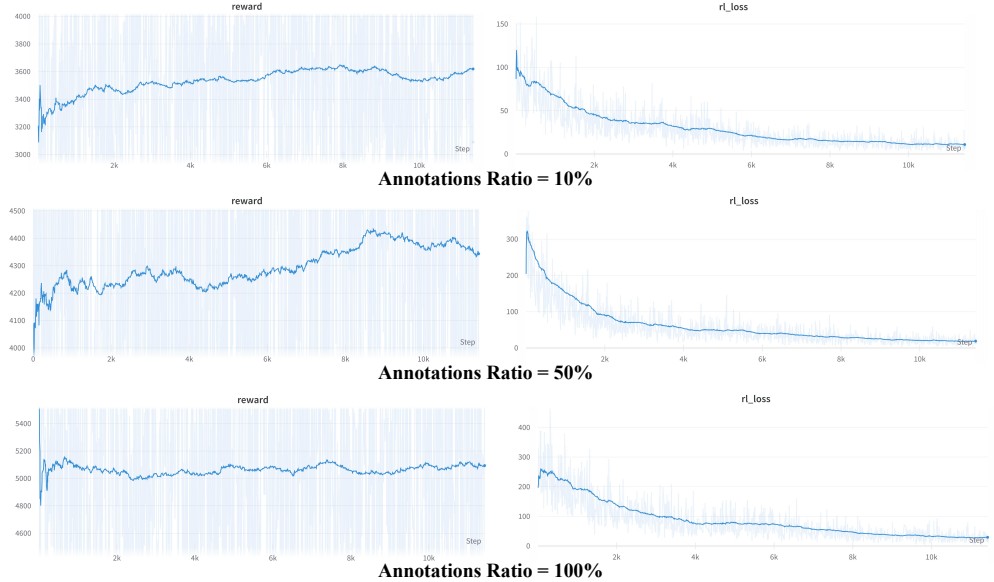

Figure 4: Train Curve.

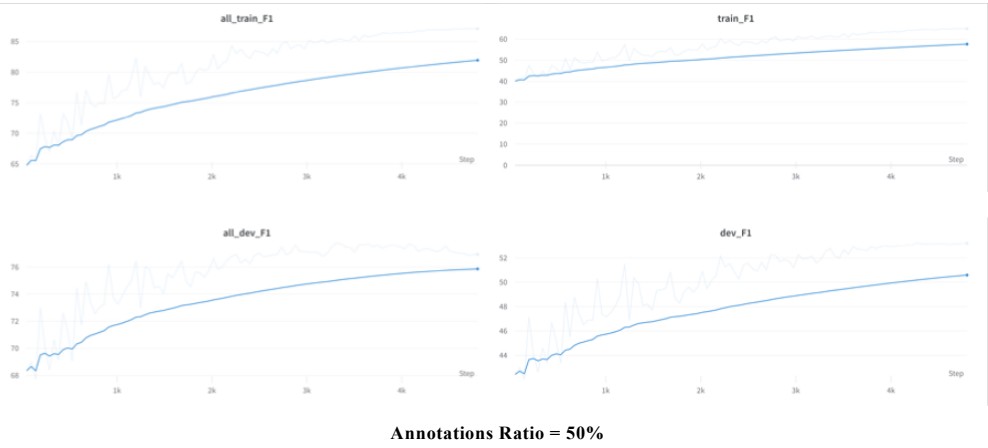

Figure 5: Metric Comparison under Ground Truth and Partially Annotations. Left: Ground Truth, Right: Partially Annotations

| Method | Data Ratio | version0 | | | version1 | | | version2 | | | average | | |
|---|---|---|---|---|---|---|---|---|---|---|---|---|---|
| | | P | R | F1 | P | R | F1 | P | R | F1 | P | R | F1 |
| DREEAM | 10% | 91.23 | 4.53 | 8.64 | 90.88 | 3.54 | 6.82 | 87.39 | 3.38 | 6.5 | 89.83 | 3.82 | 7.32 |
| | 20% | 90.74 | 10.0 | 18.02 | 92.37 | 9.65 | 17.47 | 90.86 | 9.8 | 17.69 | 91.32 | 9.82 | 17.73 |
| | 30% | 92.45 | 19.94 | 32.8 | 92.48 | 19.95 | 32.82 | 91.19 | 22.35 | 35.9 | 92.04 | 20.75 | 33.84 |
| | 40% | 93.12 | 28.08 | 43.15 | 91.92 | 34.82 | 50.51 | 92.25 | 36.34 | 52.14 | 92.43 | 33.08 | 48.6 |
| | 50% | 92.69 | 43.63 | 59.33 | 91.25 | 43.28 | 58.71 | 91.44 | 39.29 | 54.96 | 91.79 | 42.07 | 57.67 |
| | 60% | 92.16 | 48.56 | 63.6 | 90.49 | 56.52 | 69.58 | 89.52 | 55.68 | 68.66 | 90.72 | 53.59 | 67.28 |
| | 70% | 88.56 | 60.15 | 71.64 | 91.28 | 55.69 | 69.17 | 88.92 | 60.51 | 71.01 | 89.59 | 58.78 | 70.61 |
| | 80% | 87.91 | 65.18 | 74.86 | 89.83 | 62.75 | 73.89 | 86.95 | 66.49 | 75.36 | 88.23 | 64.81 | 74.7 |
| | 90% | 87.49 | 66.86 | 75.79 | 87.61 | 67.66 | 76.35 | 86.4 | 68.75 | 76.57 | 87.17 | 67.76 | 76.24 |
| Pos Weight | 10% | 84.43 | 34.1 | 48.57 | 84.61 | 43.22 | 57.21 | 85.8 | 42.21 | 56.58 | 84.95 | 39.84 | 54.12 |
| | 20% | 87.72 | 47.36 | 61.51 | 82.51 | 57.19 | 67.56 | 86.61 | 51.3 | 64.44 | 85.61 | 51.95 | 64.5 |
| | 30% | 83.57 | 61.65 | 70.95 | 87.05 | 57.04 | 68.92 | 85.75 | 59.23 | 70.07 | 85.46 | 59.31 | 69.98 |
| | 40% | 87.51 | 59.26 | 70.67 | 84.29 | 65.91 | 73.97 | 85.65 | 64.14 | 73.35 | 85.82 | 63.1 | 72.66 |
| | 50% | 83.66 | 68.09 | 75.08 | 85.78 | 66.33 | 74.81 | 85.66 | 65.92 | 74.5 | 85.03 | 66.78 | 74.8 |
| | 60% | 84.85 | 68.57 | 75.85 | 85.55 | 68.09 | 75.83 | 84.51 | 68.87 | 75.89 | 84.97 | 68.51 | 75.86 |
| | 70% | 82.77 | 73.07 | 77.62 | 83.0 | 73.13 | 77.76 | 84.37 | 71.4 | 77.34 | 83.38 | 72.53 | 77.57 |
| | 80% | 83.57 | 73.82 | 78.39 | 82.46 | 75.68 | 78.93 | 83.64 | 73.61 | 78.31 | 83.22 | 74.37 | 78.54 |
| | 90% | 83.9 | 74.54 | 78.94 | 82.48 | 76.44 | 79.35 | 82.87 | 75.8 | 79.18 | 83.08 | 75.59 | 79.16 |
| Neg Weight | 10% | 88.1 | 29.67 | 44.39 | 86.06 | 30.1 | 44.6 | 86.06 | 32.37 | 47.05 | 86.74 | 30.71 | 45.35 |
| | 20% | 82.94 | 55.7 | 66.64 | 83.72 | 55.24 | 66.56 | 85.49 | 51.25 | 64.08 | 84.05 | 54.06 | 65.76 |
| | 30% | 85.9 | 58.87 | 69.86 | 86.47 | 55.99 | 67.97 | 82.7 | 63.04 | 71.55 | 85.02 | 59.3 | 69.79 |
| | 40% | 86.1 | 62.08 | 72.14 | 85.55 | 62.72 | 72.37 | 85.19 | 64.47 | 73.39 | 85.61 | 63.09 | 72.63 |
| | 50% | 84.25 | 67.83 | 75.15 | 84.27 | 68.5 | 75.57 | 83.64 | 68.37 | 75.24 | 84.05 | 68.23 | 75.32 |
| | 60% | 84.25 | 69.76 | 76.32 | 84.39 | 69.17 | 76.02 | 82.92 | 71.29 | 76.67 | 83.85 | 70.07 | 76.34 |
| | 70% | 81.89 | 73.52 | 77.48 | 82.88 | 73.03 | 77.64 | 83.74 | 71.72 | 77.27 | 82.84 | 72.76 | 77.46 |
| | 80% | 80.99 | 76.24 | 78.54 | 82.51 | 74.83 | 78.48 | 81.58 | 74.93 | 78.11 | 81.69 | 75.33 | 78.38 |
| | 90% | 80.85 | 77.08 | 78.92 | 80.7 | 76.93 | 78.77 | 80.92 | 77.22 | 79.03 | 80.82 | 77.08 | 78.91 |
| Our PAPG | 10% | 64.47 | 72.78 | 68.37 | 62.39 | 74.98 | 68.11 | 48.65 | 83.15 | 61.39 | 58.5 | 76.97 | 65.96 |
| | 20% | 82.25 | 66.75 | 73.69 | 86.2 | 58.91 | 69.99 | 81.94 | 67.33 | 73.92 | 83.46 | 64.33 | 72.53 |
| | 30% | 83.71 | 67.98 | 75.03 | 86.03 | 63.58 | 73.12 | 80.87 | 71.56 | 75.93 | 83.54 | 67.71 | 74.69 |
| | 40% | 84.56 | 68.92 | 75.94 | 83.21 | 70.08 | 76.08 | 83.78 | 69.2 | 75.8 | 83.85 | 69.4 | 75.94 |
| | 50% | 81.4 | 72.86 | 76.89 | 80.32 | 74.34 | 77.21 | 82.55 | 73.62 | 77.83 | 81.42 | 73.61 | 77.31 |
| | 60% | 82.3 | 73.97 | 77.92 | 80.4 | 75.35 | 77.79 | 80.61 | 74.7 | 77.54 | 81.1 | 74.67 | 77.75 |
| | 70% | 83.34 | 73.57 | 78.15 | 83.27 | 73.87 | 78.29 | 83.25 | 74.36 | 78.55 | 83.29 | 73.93 | 78.33 |
| | 80% | 81.92 | 75.65 | 78.66 | 81.68 | 76.02 | 78.75 | 62.77 | 80.57 | 70.56 | 75.46 | 77.41 | 75.99 |
| | 90% | 80.83 | 77.58 | 79.18 | 80.41 | 78.01 | 79.19 | 80.97 | 77.48 | 79.19 | 80.74 | 77.69 | 79.2 |

Table 6: Results of DREEAM, Pos Weight, Neg Weight, PAPG on different ratios of annotated labels

| Sampling Ratio | version0 | | | version1 | | | version2 | | | average | | |
|---|---|---|---|---|---|---|---|---|---|---|---|---|
| | P | R | F1 | P | R | F1 | P | R | F1 | P | R | F1 |
| 0.1 | 71.21 | 82.27 | 76.34 | 77.19 | 77.3 | 77.25 | 81.53 | 71.14 | 75.98 | 76.64 | 76.9 | 76.52 |
| 0.2 | 75.22 | 79.23 | 77.17 | 79.72 | 74.82 | 77.19 | 83.26 | 68.4 | 75.14 | 79.4 | 74.15 | 76.5 |
| 0.3 | 76.79 | 77.88 | 77.33 | 80.57 | 73.6 | 76.93 | 84.88 | 66.84 | 74.79 | 80.75 | 72.77 | 76.35 |
| 0.4 | 78.26 | 77.18 | 77.72 | 81.1 | 73.58 | 77.16 | 83.94 | 67.4 | 74.77 | 81.1 | 72.72 | 76.55 |
| 0.5 | 78.35 | 77.1 | 77.72 | 83.26 | 72.1 | 77.28 | 85.23 | 65.7 | 74.2 | 82.28 | 71.63 | 76.4 |
| 0.6 | 79.54 | 76.04 | 77.75 | 83.22 | 71.41 | 76.86 | 85.41 | 65.74 | 74.29 | 82.72 | 71.06 | 76.3 |
| 0.7 | 80.74 | 75.35 | 77.95 | 83.03 | 71.12 | 76.61 | 86.41 | 64.4 | 73.8 | 83.39 | 70.29 | 76.12 |
| 0.8 | 80.43 | 75.12 | 77.68 | 83.92 | 70.6 | 76.69 | 84.65 | 65.7 | 73.98 | 83.0 | 70.47 | 76.12 |
| 0.9 | 81.04 | 74.62 | 77.7 | 84.45 | 69.83 | 76.45 | 84.44 | 65.74 | 73.92 | 83.31 | 70.06 | 76.02 |

Table 7: Action Sampling Ratio

| Data Ratio | version0 | | | version1 | | | version2 | | | average | | |
|---|---|---|---|---|---|---|---|---|---|---|---|---|
| | P | R | F1 | P | R | F1 | P | R | F1 | P | R | F1 |
| 10% | 60.69 | 74.91 | 67.06 | 57.47 | 77.17 | 65.88 | 45.89 | 84.41 | 59.46 | 54.68 | 78.83 | 64.13 |
| 20% | 81.16 | 67.2 | 73.52 | 86.34 | 58.12 | 69.47 | 83.3 | 63.88 | 72.3 | 83.6 | 63.07 | 71.76 |
| 30% | 83.1 | 66.99 | 74.18 | 83.65 | 64.82 | 73.04 | 80.95 | 69.74 | 74.93 | 82.57 | 67.18 | 74.05 |
| 40% | 85.19 | 66.35 | 74.6 | 83.25 | 69.41 | 75.7 | 83.12 | 69.13 | 75.48 | 83.85 | 68.3 | 75.26 |
| 50% | 80.44 | 72.78 | 76.42 | 78.51 | 75.1 | 76.76 | 83.24 | 72.89 | 77.28 | 80.73 | 73.59 | 76.82 |
| 60% | 83.53 | 71.8 | 77.22 | 79.9 | 74.35 | 77.02 | 80.59 | 73.6 | 76.93 | 81.34 | 73.25 | 77.06 |
| 70% | 86.14 | 69.18 | 76.74 | 87.65 | 66.63 | 75.71 | 85.57 | 69.9 | 76.94 | 86.45 | 68.57 | 76.46 |
| 80% | 85.77 | 70.19 | 77.2 | 83.64 | 72.85 | 77.87 | 79.93 | 71.16 | 75.29 | 83.11 | 71.4 | 76.79 |
| 90% | 83.86 | 74.78 | 79.06 | 82.85 | 74.42 | 78.41 | 83.77 | 74.42 | 78.82 | 83.49 | 74.54 | 78.76 |

Table 8: Value Network Performance of Our PAPG. We construct the training set three times with different random seeds, corresponding to the three versions.

| Data Ratio | nnPU | | | ImbnnPU | | | Negative Mode | | | Our PAPG | | |
|---|---|---|---|---|---|---|---|---|---|---|---|---|
| | P | R | F1 | P | R | F1 | P | R | F1 | P | R | F1 |
| 10% | 52.0 | 39.3 | 44.8 | 41.3 | 59.2 | 48.6 | 40.4 | 3.8 | 7.0 | 47.7 | 53.0 | 50.2 |
| 20% | 54.6 | 42.6 | 47.9 | 43.9 | 66.8 | 53.0 | 76.8 | 9.6 | 17.1 | 61.4 | 68.1 | 64.6 |
| 30% | 58.4 | 42.3 | 49.1 | 43.9 | 66.8 | 53.0 | 71.1 | 18.0 | 28.7 | 59.6 | 75.6 | 66.6 |
| 40% | 57.1 | 45.1 | 50.4 | 54.8 | 73.7 | 62.8 | 76.9 | 24.9 | 37.6 | 62.1 | 74.6 | 67.8 |
| 50% | 56.9 | 49.2 | 52.8 | 61.5 | 69.4 | 65.2 | 75.0 | 45.8 | 56.9 | 63.8 | 76.9 | 69.8 |
| 60% | 59.4 | 49.7 | 54.1 | 61.5 | 68.1 | 64.6 | 69.1 | 46.5 | 55.6 | 65.0 | 77.7 | 70.8 |
| 70% | 61.7 | 51.5 | 56.1 | 62.6 | 67.4 | 64.9 | 82.2 | 52.7 | 64.2 | 72.6 | 77.7 | 75.1 |
| 80% | 63.0 | 52.7 | 57.4 | 63.2 | 72.8 | 67.7 | 79.5 | 64.4 | 71.2 | 78.9 | 73.0 | 75.8 |
| 90% | 70.4 | 47.5 | 56.7 | 62.7 | 77.2 | 69.2 | 82.5 | 69.1 | 75.2 | 75.2 | 78.7 | 76.9 |

Table 9: The results of CIFAR10 dataset. We consider the original class 'airplane' as the positive targets.

| Data Ratio | Pos Weight | | | Neg Weight | | | Negative Mode | | | Our PAPG | | |
|---|---|---|---|---|---|---|---|---|---|---|---|---|
| | P | R | F1 | P | R | F1 | P | R | F1 | P | R | F1 |
| 20% | 72.8 | 69.5 | 71.1 | 89.7 | 39.5 | 54.9 | 87.9 | 9.9 | 17.9 | 79.4 | 67.2 | 72.8 |
| 40% | 74.3 | 75.0 | 74.7 | 82.5 | 65.6 | 73.1 | 90.3 | 28.0 | 42.8 | 83.0 | 74.4 | 78.5 |
| 60% | 74.6 | 79.0 | 76.7 | 80.1 | 74.0 | 76.9 | 96.0 | 47.0 | 63.1 | 79.4 | 76.5 | 77.9 |
| 80% | 72.5 | 83.1 | 77.4 | 83.1 | 75.7 | 79.2 | 92.8 | 65.3 | 76.6 | 82.4 | 77.5 | 79.9 |

Table 10: The results of other annotation ratios on MS-COCO dataset.

| Standard Deviation | | | | | | | | | |
|---|---|---|---|---|---|---|---|---|---|
| 10% | | 30% | | 50% | | 70% | | 90% | |
| F1 | mAP | F1 | mAP | F1 | mAP | F1 | mAP | F1 | mAP |
| 68.3(0.12) | 66.6(0.33) | 77.0(0.30) | 77.5(0.25) | 79.1(0.15) | 80.4(0.13) | 79.0(0.10) | 81.4(0.15) | 80.5(0.05) | 83.4(0.05) |

Table 11: The standard deviations ($\cdot$) of F1 and mAP were computed from three different runs on the MS-COCO dataset.

Table 12: Experimental Results on COCO datasets with varying ratios of positive classes annotations.

| Method | 10% | | 30% | | 50% | | 70% | | 90% | |
|---|---|---|---|---|---|---|---|---|---|---|
| | F1 | mAP | F1 | mAP | F1 | mAP | F1 | mAP | F1 | mAP |
| ERP | - | 63.8 | - | 71.0 | - | 73.5 | - | 73.8 | - | 74.4 |
| ROLE | - | 58.2 | - | 72.4 | - | 76.6 | - | 79.5 | - | 81.1 |
| P-ASL+Negative | 45.2 | 66.9 | 52.1 | 74.6 | 54.0 | 76.9 | 71.9 | 81.0 | 80.3 | 83.3 |
| P-ASL+Counting | 5.1 | 46.4 | 26.4 | 63.4 | 53.7 | 76.1 | 71.6 | 80.1 | 60.4 | 80.4 |
| Pos Weight | 66.7 | 64.3 | 73.0 | 72.7 | 75.7 | 76.7 | 76.0 | 79.9 | 77.5 | 82.6 |
| Neg Weight | 24.0 | 56.9 | 68.7 | 74.8 | 75.9 | 78.0 | 77.9 | 79.7 | 80.5 | 82.8 |
| Negative Mode | 6.4 | 50.6 | 33.7 | 64.3 | 52.9 | 73.8 | 72.3 | 81.2 | 80.1 | 83.5 |
| PAPG (Ours) | 68.3 | 66.6 | 77.0 | 77.5 | 79.1 | 80.4 | 79.0 | 81.4 | 80.5 | 83.4 |

| Item | Content or Triples |
|---|---|
| **Title** | Guido Bonatti |
| **Document** | Guido Bonatti (died between 1296 and 1300) was an Italian mathematician, astronomer and astrologer, who was the most celebrated astrologer of the 13th century. Bonatti was advisor of Frederick II, Holy Roman Emperor, Ezzelino da Romano III, Guido Novello da Polenta and Guido I da Montefeltro. He also served the communal governments of Florence, Siena and Forlì. His employers were all Ghibellines (supporters of the Holy Roman Emperor), who were in conflict with the Guelphs (supporters of the Pope), and all were excommunicated at some time or another. Bonatti 's astrological reputation was also criticised in Dante's Divine Comedy, where he is depicted as residing in hell as punishment for his astrology. His most famous work was his Liber Astronomiae or 'Book of Astronomy', written around 1277. This remained a classic astrology textbook for two centuries. |
| **DREEAM** | ⟨Dante, notable work, Divine Comedy⟩ |
| **Pos Weight** | ⟨Dante, notable work, Divine Comedy⟩
⟨Divine Comedy, creator, Dante⟩
⟨Divine Comedy, author, Dante⟩ |
| **Neg Weight** | ⟨Guido Bonatti, notable work, Liber Astronomiae⟩
⟨Guido Bonatti, notable work, Book of Astronomy⟩
⟨Dante, notable work, Divine Comedy⟩
⟨Divine Comedy, author, Dante⟩ |
| **Our PAPG** | ⟨Guido Bonatti, notable work, Liber Astronomiae⟩
⟨Guido Bonatti, notable work, Book of Astronomy⟩
⟨Dante, notable work, Divine Comedy⟩
⟨Divine Comedy, creator, Dante⟩
⟨Divine Comedy, author, Dante⟩
⟨Liber Astronomiae, author, Guido Bonatti⟩ |
| **Ground Truth** | ⟨Guido Bonatti, date of death, 1296⟩
⟨Guido Bonatti, date of death, 1300⟩
⟨Divine Comedy, characters, Guido Bonatti⟩
⟨Divine Comedy, creator, Dante⟩
⟨Divine Comedy, author, Dante⟩
⟨Book of Astronomy, author, Guido Bonatti⟩
⟨Liber Astronomiae, author, Guido Bonatti⟩
⟨Guido Bonatti, country of citizenship, Italian⟩
⟨Guido Bonatti, notable work, Liber Astronomiae⟩
⟨Dante, notable work, Divine Comedy⟩
⟨Guido Bonatti, present in work, Divine Comedy⟩
⟨Guido Bonatti, notable work, Book of Astronomy⟩ |

Table 13: An Example from Re-DocRED