# OpenReview forum: "Make Small Data Great Again: Learning from Partially Annotated Data via Policy Gradient for Multi-Label Classification Tasks"
_ICLR.cc/2024/Conference — ICLR 2024 Conference Withdrawn Submission_

### Official Review · Reviewer_svcR · 2023-10-16

**Soundness:** 2 fair
**Presentation:** 2 fair
**Contribution:** 2 fair
**Rating:** 3
**Confidence:** 4

**Summary:**

In this submission, authors investigate the understudied problem of partially annotated multi-label classifcation, where  only a subset of positive classes is annotated. To deal with this problem, authors propose a new method named Partially Annotated reinforcement learning with a Policy Gradient algorithm (PAPG) which can overcome the challenges associated with a scarcity of positive annotations and severe label imbalance.

**Strengths:**

1. The problem, partially annotated multi-label classifcation, is novel.

2. A novel method named PAPG is proposed, which can overcome the challenges associated with a scarcity of positive annotations and severe label imbalance.

3. Experiments validate the effectiveness of the proposed PAPG method.

**Weaknesses:**

1. The motivation might be unreasonable. As stated in abstract, "This task encounters challenges associated with a scarcity of positive annotations and severe label imbalance". These challenges are indeed problem to deal with, but in my opinion, the main challenges correspond to that the negative samples are unknown or we do not know which are negative samples among the remaining unlabeled samples. In other words, the main challenges are the same as PU learning.

2. For the two challenges mentioned in this paper, i.e., a scarcity of positive annotations and severe label imbalance, it is unclear how can the proposed PAPG method overcome them.

3. As stated in Introduction, "Consequently, many advanced methods of PU learning (Su et al., 2021; Acharya et al., 2022; Luo et al., 2021) cannot readily adapt to our multi-label settings". In my opinion, if we focus each label one by one, it is a PU learning problem. Thus, the partially annotated multi-label classifcation problem can be solved via binary relevance strategy where the base classifier is trained with any off-the-shelf PU learning methods.

4. For multi-label classification, it is very important to model the correlations among labels. But it is unclear how can the proposed PAPG method model label correlations.

5. The writtting can be greatly improved. For exampe:
(1) The title is inappropriate.
(2) Errors exist in notations (especially the first paragraph in section 3.1).
(3) What does RARL mean in the last sentence of this paper?

**Questions:**

If author disagree with my comments in Weaknesses, please clarify them in the rebuttal phase.

---

> ### Author Response · Authors · 2023-11-15
>
> Thank you very much for your thoughtful feedback and acknowledgment of our `novelty and effectiveness`.
>
> > (W.1) The motivation might be unreasonable. As stated in the abstract, "This task encounters challenges associated with a scarcity of positive annotations and severe label imbalance". These challenges are indeed problem to deal with, but in my opinion, the main challenges correspond to that the negative samples are unknown or we do not know which are negative samples among the remaining unlabeled samples. In other words, the main challenges are the same as PU learning.
>
>
> **A:** We agree that we have the same challenge of unknown negative samples as PU learning. Such a challenge is **commonly faced by partially annotated problems**. Besides, we want to emphasize that solving multi-label classification tasks with partial annotation also **encounters more severe difficulties than PU learning**:
> - positive-unknown imbalance;
> - unknown prior positive-negative distribution or actual ratio of positive missing.
>
> PU learning generally **supposes an actual balanced positive-negatvie distribution or given the prior positive-negative distribution**. Su et al. [1] puts forward doubts on the strong assumption and has explored **PU learning from imbalanced data**. Their setting in the PU learning scenario is the most related to us, and **we have conducted experiments for comparison**.
>
> Our task setting can be considered as a more general form of PU learning and can be applied to more realistic task challenges, such as relation extraction and image classification in NLP and CV.
>
>
> > (W.2) For the two challenges mentioned in this paper, i.e., a scarcity of positive annotations and severe label imbalance, it is unclear how can the proposed PAPG method overcome them.
>
> **A:** Our method addresses the above two challenges from two carefully designed mechanisms.
>
> - **Self-supervised data enhancement. -> Scarcity positive labels**:
> We produce pseudo positives with consideration of both policy and value networks to guarantee pseudo positive classes with high confidence.
> - **Designing local rewards provided by value network and global rewards -> label imbalance**:
> Our value network **exploits data distribution** with high-quality label augmentation; our policy network **performs exploration** guided by the value network. **The global reward can encourage our policy network to predict a greater number of classes as TRUE**, and the local rewards monitor the exploration of our policy network so as not to stray too far from the existing annotations.
>
>
>
> > (W.3) As stated in Introduction, "Consequently, many advanced methods of PU learning (Su et al., 2021; Acharya et al., 2022; Luo et al., 2021) cannot readily adapt to our multi-label settings". In my opinion, if we focus each label one by one, it is a PU learning problem. Thus, the partially annotated multi-label classifcation problem can be solved via binary relevance strategy where the base classifier is trained with any off-the-shelf PU learning methods.
>
> **A:** Thanks for your thoughtful comments. Actually, we leverage binary relevance as the base of our framework. However, **many advanced PU learning methods *that suppose a given positive-negative distribution prior or assume a balanced positive-negative ratio* are not adapted for our settings**.
>
> We want to emphasize the challenges of multi-label classification beyond those of PU learning:
> - **the severe positive-negative imbalance** and **missing labels exacerbate the imbalance and plague recognizing positives**;
> - **unknown distribution prior**.
>
>
> Thus, our task setting can be considered as **a more general form of PU learning**, and our framework **can be adapted for the PU learning setting**. Therefore, we conducted experiments on the PU learning setting and **compared to nnPU and ImbalancednnPI methods**, a method for general PU learning and a specific method explored PU learning from imbalanced data. Compared to them, **our framework also has superiority in the PU learning setting** (demonstrated in Table 1 in experiment 1).
>
>
> [1] Su et al., Positive-unlabeled learning from imbalanced data, IJCAI 2021.

---

> ### Author Response · Authors · 2023-11-15
>
> > (W.4) For multi-label classification, it is very important to model the correlations among labels. But it is unclear how can the proposed PAPG method model label correlations.
>
>
> **A:** Thank you for pointing this out. Indeed, we have considered the label correlations in our challenge. However, in our setting, label correlations may bring correlation bias to the severe scarcity of positive labels. Therefore, we did not explicitly model the correlations in our model. **A similar statement is also indicated in Huang et al. [2]**: "Label correlation can be exploited to improve the performance of multi-label classifiers, and it is usually learned or calculated from the label matrix of the training data. **But an inaccurate result of label correlation will be obtained if part of ground-truth labels is missing**" and "**Inaccurate decision function of a multi-label classifier will be made if** the instances without one class label are simply treated as negative instances or **a multi-label classifier tries to exploit label correlation from the incomplete label matrix directly**".
>
> Nevertheless, we do appreciate the suggestion and would like to explore the potential of leveraging the label correlations to enhance our framework in future work.
>
> > (W.5) The writing can be greatly improved. For example: (1) The title is inappropriate. (2) Errors exist in notations (especially the first paragraph in section 3.1). (3) What does RARL mean in the last sentence of this paper?
>
> **A:** Thank you for your careful review. We have:
> - revised the title to `Learning from Partially Annotated Data via Policy Gradient for Multi-Label Classification Tasks`;
> - carefully refined the notations in Section 3.1 and Algorithm 1;
> - fixed "RARL" to "PAPG".
>
> Sorry about the typos; we have carefully reviewed our paper and fixed them in our revised version.
>
> [2] Huang et al., Improving multi-label classification with missing labels by learning label-specific features, Information Sciences 2019.

---

### Official Review · Reviewer_Ab3u · 2023-10-19

**Soundness:** 2 fair
**Presentation:** 2 fair
**Contribution:** 2 fair
**Rating:** 5
**Confidence:** 3

**Summary:**

This paper addresses the problem of partially annotated multi-label classification, where only a subset of positive classes is annotated, leading to imbalanced and challenging learning scenarios. The proposed idea exploits reinforcement learning and designs local rewards assessed by a value network and global rewards assessed by recall functions to mitigate class imbalance issues. The proposed approach is evaluated across various classification tasks and demonstrates its effectiveness in improving upon previous methods.

**Strengths:**

- Multi-label annotations are challenging. Partial labeling is a great way to alleviate labeling overheads.

- The approach introduced in the paper is straightforward, aligning with intuitive problem-solving strategies. It offers a clear and understandable solution to the challenges at hand.

**Weaknesses:**

- The literature survey for weakly supervised learning in this paper is incomplete. The survey only covers some settings under multi-class single-label scanrio and misses the weak supervion under multi-label learning, such as missing-label learning. This miss is critical as the major novelty of this paper comes from proposing method in this setting. The comparisions with such methods are also missing, so that the effectiveness of the proposed method is not well supported. The baselines are considerably inadequate, and the results on some famous multi-label classification dataset, such as NUS-WIDE dataset, are missing.

- The proposed method itself is simple and straightforward. Therefore, it would be better to further analyze the design choices of the proposed model to claim the impact of the proposed method. For example, why is the loss function used? Is there any better option for the authors to try for the loss function?

**Questions:**

It would be better to add several more results in the main paper, and the novelty is debatable.

---

> ### Author Response · Authors · 2023-11-15
>
> We sincerely thank you for your constructive comments and acknowledgment of our work.
>
> > (W.1) The literature survey for weakly supervised learning in this paper is incomplete. The survey only covers some settings under multi-class single-label scenarios and misses the weak supervision under multi-label learning, such as missing-label learning. This miss is critical as the major novelty of this paper comes from proposing a method in this setting. The comparisions with such methods are also missing, so that the effectiveness of the proposed method is not well supported. The baselines are considerably inadequate, and the results on some famous multi-label classification datasets, such as NUS-WIDE dataset, are missing.
>
> **A:** Thanks for your constructive comments.
>
> **For the literature survey for multi-label learning**, we have noticed four settings of multi-label learning with weak supervision relevant to our setting.
> - POL refers to partially observed labels, which supposes that partial annotations contain both positive and negative classes, such as [1].
> - PPL refers to partially observed positive labels, which supposes that only positive classes are annotated but requires that there is at least one positive label per instance, such as [3].
> - SPL refers to a single positive label, which supposes that each instance only has one positive label, such as [2].
> - PML refers to partial multi-label learning, which supposes that each instance has a set of candidate labels where the number of positive labels is unknown, such as [4].
>
> We focus on the partially observed setting with random ratio of positive class annotations and no negative class annotation, which is different from the above settings. It is worth specifically pointing out that we do not follow the PPL setting with promising at least one positive label per instance. Because in some tasks, such as relation extraction, the instance may not have any positive classes.
>
>
> **For comparisons with partially observed multi-label learning,** we verify the advantage of our method in **two domains**:
> - **In the Nature Language Process domain**, we conducted experiments in the relation extraction task and compared our method with some simple baselines, **because there are no models designed for partially annotated settings in relation extraction**.
> - **In the Computer Vision domain**, we conducted an experiment in the multi-label image classification task. We **have compared to previous work designed for POL setting** [1] by **re-running their official code on our datasets**. We provide **comparisons (MAP) with additional previous methods** that provides code in the following Table.
>
> |MAP score|10%|30%|50%|70%|90%|
> |---|---|---|---|---|---|
> |EPR[2]|63.8|71.0|73.5|73.8|74.4|
> |ROLE[2]|58.2|72.4|76.6|79.5|81.3|
> |PAPG(Ours)|66.6|77.5|80.4|81.4|83.4|
>
> **For experiments on more multi-label classification datasets, such as NUS-WIDE dataset,** to prove the effectiveness and generalizability of our framework, we **conduct experiments with tasks in different domains, i.e., PU learning, NLP, and CV**. Thus we only **choose one representative multi-label image classification dataset**. We will add more experiments on different datasets.
>
>
>
>
> > (W.2) The proposed method itself is simple and straightforward. Therefore, it would be better to further analyze the design choices of the proposed model to claim the impact of the proposed method. For example, why is the loss function used? Is there any better option for the authors to try for the loss function?
>
> **A:** We have conducted an ablation and variant study in Section 4.4, which **includes analysis of different reward designs and different training strategies**. For example, we show the advantages of both local reward and global reward we designed, and we compared our framework with directly supervised self-training to show the **benefits of RL module besides the benefits of pseudo positive classes augmentation**, etc.
>
> [1] Ben-Baruch et al., Multi-label classification with partial annotations using class-aware selective loss, CVPR 2022.
>
> [2] Cole et al., Multi-label learning from single positive labels, CVPR 2021.
>
> [3] Zhang et al., Simple and robust loss design for multi-label learning with missing labels, arXiv preprint arXiv:2112.07368, 2021.
>
> [4] Xie et al., Partial Multi-Label Learning, AAAI 2018.

---

> ### Author Response · Authors · 2023-11-15
>
> > (Q1.) The novelty is debatable.
>
> **A:** We want to underline our novelty:
> - To the best of our knowledge, **applying RL framework in a partially annotated *multi-label* classification problem** is entirely new.
> - We do not simply apply traditional RL methods for our setting. Instead, we **specifically design novel reward functions and training strategies** for the partially annotated challenge. The ablation study shows the effectiveness of our designs.
> - We combine supervised learning with label augmentation and RL training to **take advantage of both exploitation and exploration abilities**, which is necessary when lacking positive labels.
> - We demonstrate **consistent improvements on three tasks in different domains**, i.e., a PU learning setting, an NLP task, and a CV task, which shows **the robustness and generalizability of our framework**.

---

### Official Review · Reviewer_QJLy · 2023-10-22

**Soundness:** 3 good
**Presentation:** 3 good
**Contribution:** 3 good
**Rating:** 6
**Confidence:** 4

**Summary:**

The proposed methods explore a new approach on partially annotated multi-label classification by using RL-based framework. The approach contains a local rewards assessed by a value network and global rewards assessed by recall functions to guide the learning of policy network. The experiments on binary image classification, multi-label image classification, and document-level relation extraction show the effectiveness of the proposed method.

**Strengths:**

+ The proposed methods explore a pretty interesting direction (RL based design) on partially annotated multi-label classification.

+ the paper is well-organized and have clear figures and demonstrations.

+ The paper is technically sound and provide necessary analysis.

+ The author test the algorithm on multiple tasks.

**Weaknesses:**

- usually the algorithms contain the DRL methods will have a relatively higher variance on the performance. May I know what is the variance of your model's F1 and mAP in multi-label image classification?

- How many time did you repeat your experiments?

**Questions:**

- How large is the computation overhead in algorithm compared with the standard supervised learning?

- What will be the obstacles if you run this algorithm on a very large image dataset compared to the standard supervised learning? For example, 9 million partially annotated training images (OpenImages V6)?

- How many time did you repeat your experiments? May I know what is the variance of your model's F1 and mAP in multi-label image classification?

- It would be better if you can show a figure reflecting the training process, e.g., x-axis: training steps, y-axis: accumulated reward or mAP.

---

> ### Author Response · Authors · 2023-11-15
>
> We sincerely appreciate you for your careful review and insightful comments. We provide detailed replies to your comments and hope we can resolve your major concerns.
>
> > (W.1) usually the algorithms contain the DRL methods will have a relatively higher variance on the performance. May I know what is the variance of your model's F1 and mAP in multi-label image classification?
> >
> > How many time did you repeat your experiments? May I know what is the variance of your model's F1 and mAP in multi-label image classification?
>
> **A:** Great question. We indeed considered two different scenarios to evaluate the robustness of our method.
> - **Different dataset versions:**
> To simulate partial annotation, we randomly reserve a ratio of positive classes and treat other classes as unknown. To fully verify the effectiveness and stability of our model, we **randomly constructed three versions of data sets** (refer to **Appendix B** **Table 6** for detailed results on DocRE). It can be seen that our method achieves consistent improvement in all the dataset versions.
> - **Different runs in the same dataset:**
> For multi-label image classification, we repeat three runs in the same dataset. The **standard deviations of F1/MAP scores of Table 2 in paper** is as follows.
>
> |10%|30%|50%|70%|90%|
> |---|---|---|---|---|
> |0.12/0.33|0.30/0.25|0.15/0.13|0.10/0.15|0.05/0.05|
>
> Thank you for pointing out. We have updated the above results in our revised paper.
>
>
>
>
> > (Q1) How large is the computation overhead in algorithm compared with the standard supervised learning?
>
> **A:** We first want to clarify the relations between our methods and standard supervised learning methods.
> - Our framework is applicable to be built upon any model structures with supervised learning.
> - We used existing SOTA-supervised learning models as our policy and value networks.
> - We supervised train our policy network.
>
>
> **In the training process:**
> Our framework consists of a value network and a policy network, which leads to **a doubling of parameters and training time** compared to standard supervised learning.
>
>
>
> **In the inference process:**
> We only leverage the policy network to decide $\texttt{TRUE}$ classes. Thus, **the inference computation is equal to the standard supervised learning methods** after geting a well-trained model.
>
>
> Since our framework can be treated as a teacher-student architecture, we would like to **explore the potential of a lightweight yet high-performance policy model** in future work.
>
>
>
> > (Q2.) What will be the obstacles if you run this algorithm on a very large image dataset compared to the standard supervised learning? For example, 9 million partially annotated training images (OpenImages V6)?
>
> **A:** Our framework is **model-agnostic and acts as a plug-in strategy**, so there are no problems to apply our framework for other datasets. However, there is no denying the fact that we utilize an extra value network to assist learning. It requires twice the storage and training time compared with standard supervised learning and thus needs more space and time computation on large datasets.
>
> **For the reasons we did not experiment on other datasets**, the baseline model we chose for multi-label image classification is Ben-Baruch et al. [1], where they reported results on MS-COCO, OpenImages V6, and LVIS. But the code base provided by the author only contains the training code on MS-COCO and is not adapted to OpenImage V6.
>
> Moreover, we **conduct experiments with tasks in different fields, i.e., PU learning, NLP, and CV.** Thus, we only **choose one representive multi-label image classification** dataset MS-COCO.
>
>
> > (Q3.) It would be better if you can show a figure reflecting the training process, e.g., x-axis: training steps, y-axis: accumulated reward or mAP.
>
> **A:** Thanks for pointing out. Indeed, we **have provided training curves** during *submission* in **Appendix B.1** of supplementary material. Specifically, in **Figure 4**, we display the reward and loss curves of our model in three annotation rations: 10%, 50%, and 100%. We would like to add more details if there are further concerns.
>
> [1] Ben-Baruch et al., Multi-label classification with partial annotations using class-aware selective loss, CVPR 2022.

---

> > ### Comment · Reviewer_QJLy · 2023-11-22
> >
> > Thanks for the reply! Authors have mostly addressed my concerns.

---

> > > ### Author Response · Authors · 2023-11-22
> > > **Thank you for confirmation**
> > >
> > > Dear Reviewer QJLy,
> > >
> > > We appreciate your effort in reading our rebuttal and confirming that we have mostly addressed your concerns!
> > >
> > > Authors

---

### Official Review · Reviewer_LdF2 · 2023-10-31

**Soundness:** 2 fair
**Presentation:** 3 good
**Contribution:** 2 fair
**Rating:** 3
**Confidence:** 4

**Summary:**

This paper proposes the PAPG framework to deal with the partially annotated multi-label classification.
It first discussed the partially annotated learning in the multi-label classification tasks, which bears great significance.
And then proposed the local and global rewards which is the main contribution of this paper.
With an iterative training strategy, PAPG gains good results.

**Strengths:**

1. This paper touches on the partially annotated multi-label classification, which is an area of great significance.
2. This paper has some experiments to support its conclusions.
3. This paper is easy to follow, especially the descriptions of the modelling process of RL in Sec. 3.2, which is very clear.
4. The proposed PAPG gains promising results on multiple benchmarks.

**Weaknesses:**

1. Methods do not support assertions.
This paper claims the local reward is the immediate reward in Sec. 1. and it provides immediate value estimation of each action in Sec. 3.2
However,  these immediate rewards are summed up in dimension $C$ to get the final reward according to Eq. 3.
It does not have immediate properties.
It has the same frequency as the global reward.
If this problem is not well explained, then this problem will be fatal.
I see no reason to use RL for this task.

2. Poor formula expression. It is shown in "Question" part of my review.

**Questions:**

1. Whether $\theta$ or $\theta^*$ is used in getting $\overline{\mathcal{y}}$ in line13 of Algorithm 1.
2. I am confused with this formula, $\hat{y}^c_i=V_\lambda(x_i)_c=1$. Please explain the process and meaning of it.
3. I think the dataset for the training value network is $(X, Y, \overline{Y})$, not $(X, Y \cup  \overline{Y})$, according to Eq. 4.
It uses $ Y$ and $\overline{Y})$ , not $Y \cup  \overline{Y}$.

---

> ### Author Response · Authors · 2023-11-15
>
> Thank you very much for your constructive comments and suggestions.
>
> > (W.1) Methods do not support assertions. This paper claims the local reward is the immediate reward in Sec. 1. and it provides immediate value estimation of each action in Sec. 3.2 However, these immediate rewards are summed up in dimension to get the final reward according to Eq. 3. It does not have immediate properties. It has the same frequency as the global reward. If this problem is not well explained, then this problem will be fatal.
>
> **A:** Thank you for the careful comment. We want to clarify our rewards design by following aspects:
>
> - As formulated in Section 3.2, we formalize the partially annotated multi-label prediction as a **one-step** MDP problem. In our reward design, the terms "local" and "global" are both used to characterize the "goodness" of a prediction given an input (i.e., a state-action pair). Therefore, they are both immediate rewards in the considered RL framework.
> - By **local reward**, we refer to the reward assigned to the prediction of each class $y^c_i$. The underlying insight of such design has been stated in the Rewards part of paper: " `Intuitively, the local rewards offer a preliminary yet instructive signal to guide the learning process in our PAPG framework. This signal inherits the *exploitation* aspect from the supervised loss training (as the value network is trained through supervised learning). Its purpose is to prevent the PAPG from engaging in excessively invalid exploratory behavior within the large action space, thereby enhancing the overall learning efficiency.`"
> - By **global reward**, we refer to the reward assigned to the overall prediction $\mathbf{y}_i=[y^0_i, ..., y^{|C|}_i]$ of a data sample (i.e. a state $\mathbf{x}_i$), which "`stimulate more comprehensive exploration during the learning process`" and "`mitigates distribution bias and encourages our policy network to predict a greater number of classes as TRUE`.
>
> Sorry about any confusion that we brought up here. We have revised our manuscript with further clarification in **Introduction** as the following paragraph:
> "`Specifically, we combine the exploration ability of reinforcement learning and the exploitation ability of supervised learning by designing a policy network (as a multi-label classifier) learning with a policy-based RL algorithm and a value network (as a critic) trained with a supervised loss to provide local reward. Besides, beyond the local reward, we design a global reward assessing predictions of each instance, which contributes most to alleviating the imbalanced problem.`"
>
>
> > (W.1) I see no reason to use RL for this task.
>
> **A:** We summarize the main motivation for using RL for our task:
>
> - **Inherent data balancing:** As stated in **Introduction**, the traditional approach that uses weakly supervised learning (WSL) methods for multi-label classification tasks struggles with predicting of **long-tail classes** and leads to a `performance drops significantly when the ratio of annotated positive classes decreases`. Different from WSL, which focuses on exploiting the original incomplete data distribution, the exploration ability of RL **has advantage of finding more positive classes when lacking positive annotations**, and **avoids the overfitting problem that supervised learning generally faces, especially when the observed data distribution is biased**.
> - **Related Successful RL attempts:** RL has been demonstrated as a successful learning approach in many weakly supervised settings (refer to **Related Works Reinforcement Learning under Weak Supervision** for details). Specifically, Luo et al. [1] has shown effective attempts by applying the RL framework to a simple PU learning setting. In our task, we inherit from a similar motivation to solve a more complicated and realistic task setting.
> - **Experimental Effectiveness:**  Empirically, as comprehensively presented in our **Experiments** Section, we show that our proposed RL framework, with careful reward designs and training strategy, **has shown a consistent improvement across three tasks from different domains**, especially when it comes to severe data imbalanced and extremely partial annotation problem. Refer to Table 1-3, Figure 1, Figure 3, for exemplar results.
>
> [1] Luo et al., Pulns: Positive-unlabeled learning with effective negative sample selector. AAAI 2021.

---

> > ### Author Response · Authors · 2023-11-15
> >
> > > (Q.1) Whether $\theta$ or $\theta^*$ is used in getting $\bar{\mathcal{y}}$ in line13 of Algorithm 1.
> >
> > **A:** We used $\theta$ rather than $\theta^*$ in line 13 of Algorithm 1. There are two reasons:
> > - In early training, because our validation set is also partially annotated, the best model (with parameter $\theta^*$) selected based on validation performance is likely to predict a small number of positive classes. However, we want more high-confidence pseudo-positives in the early training stage.
> > - In later training, the policy and value networks are nearly convergent; the pseudo-positive set may not change if leveraging $\theta^*$. We use the policy model with $\theta$ to take more exploration, thereby reducing the likelihood of the model falling into local optimal in the later training stage.
> >
> >
> > > (Q.2) I am confused with this formula, $\hat{y}_i^c =V_λ(\mathbf{x}_i)_c = 1$. Please explain the process and meaning of it.
> >
> > **A:**  The process and meaning are that:
> > - $\hat{y}_i^c = V_λ(\mathbf{x}_i)_c$ in {0, 1} refers to the predition of value network. $\hat{y}_i^c = V_λ(\mathbf{x}_i)_c = 1$ refers to the prediction of class $c$ that value network outputs for the sample $\mathbf{x}_i$ being $\texttt{TRUE}$.
> > - $\pi_\theta (\hat{y}_i^c =1| \mathbf{x}_i)$ outputs the probability of prediction ($\hat{y}_i^c = 1$).
> > - We update $\bar{\mathcal{Y}}$ with consideration of both policy and value networks. The underlying heuristic is that such a process can produce pseudo-positives with high confidence.
> >
> > We have added an explanation in Section 3.3 for clarification.
> >
> >
> > > (Q.3) I think the dataset for the training value network is $(X, Y, \bar{Y})$, not $(X, Y\cup\bar{Y})$, according to Eq. 4. It uses $Y$ and $\bar{Y}$, not $Y\cup\bar{Y}$.
> >
> > **A:** Thanks for pointing out it.  We have refined our Algorithm 1. Please refer to the revised paper for more details.

---

> > ### Comment · Reviewer_LdF2 · 2023-11-22
> > **Response to Authors**
> >
> > Thanks to the author for their careful responses, which partially answered my confusion.
> >
> > However, after reading the answer, I still don't agree with the necessity of using RL modeling for this task.
> > I hope the author will do more work on the motivation of modeling with RL in this task.

---

> > > ### Author Response · Authors · 2023-11-22
> > > **Reply to 'Response to Authors'**
> > >
> > > Dear Reviewer,
> > >
> > > Thank you for reading our rebuttal and providing follow-up comments. Please note that we did **not** claim there is a  `necessity of using RL modeling for this task`. Nevertheless, we demonstrate a superior performance and essentially new framework by incorporating RL to this task. Besides the reasons of using RL that we put in our [last reponse](https://openreview.net/forum?id=bUGzjiUsIq&noteId=VxDoVd3cbE) (in terms of task definition, prior insights, experimental efficacy), we would also like to point out that:
> > >
> > > - Reviewer QJLy considers our framework as **a pretty interesting direction (RL based design) on partially annotated multi-label classification** and **technically sound**.
> > > - Reviewer Ab3u considers our framework as **straightforward, aligning with intuitive problem-solving strategies** and **offers a clear and understandable solution **
> > > - Reviewer svcR considers our framework as  **novel**
> > >
> > > Indeed, we put many effort into improving the **motivation** part of this work (check the **Introduction section of the revised submission**). Please kindly let us know your detailed concerns or suggestions and we would like to try our best to answer and improve this work.
> > >
> > > Authors

---

### Author Response · Authors · 2023-11-15
**Response to all reviewers**

Dear reviewers,

We appreciate the efforts of all reviewers and your constructive comments! We highlight the main strengths and contributions of our work as follows:

- **Significant Problem**. We study the partially annotated multi-label classification problem, "which is an area of great significance" (**Reviewer LdF2**); "Multi-label annotations are challenging. Partial labeling is a great way to alleviate labeling overheads" (**Reviewer Ab3u**); and "The problem, partially annotated multi-label classification, is novel." (**Reviewer svcR**).
- **Novel Mothed**. To solve the challenges of partially annotated multi-label classification, we propose a "novel" (**Reviewer svcR**) method combining the exploration ability of reinforcement learning and the exploitation ability of supervised learning, which is "straightforward, aligning with intuitive problem-solving strategies, offering a clear and understandable solution to the challenges at hand." (**Reviewer Ab3u**). With specifically designed reward functions and training strategy, our framework is "technically sound" (**Reviewer QJLy**) and can overcome the challenges associated with a scarcity of positive annotations and severe label imbalance." (**Reviewer svcR**).
- **Convincing Experiments**. To prove our method's generalizability and effectiveness, we conduct experiments on three different tasks of different domains. Our experiments "gain promising results on multiple benchmarks." **(Reviewer LdF2)** and "validate the effectiveness of the proposed PAPG method." **(Reviewer svcR)**. Moreover, we provide "necessary analysis" **(Reviewer QJLy)** and have "experiments to support conclusions." **(Reviewer LdF2)**.

We've revised our manuscript according to the reviewers' suggestion (highlighted in orange in the uploaded revision pdf). Detailed responses to each reviewer's concerns are carefully addressed point-by-point. Below summarize the major updates we've made:

- clarify and refine the notation in Section 3.1 and Algorithm 1.

- analyze more related work in the Introduction.

- compare to more models on the COCO dataset.

We believe our proposed method has feasibility and applicability to many reality multi-label classification tasks with difficulties in achieving complete annotations. We provide comprehensive responses below to address each reviewer's concern and are eager to participate in further discussions if there are additional comments.

Best,

Authors